# Biomarkers of Intestinal Injury in Colic

**DOI:** 10.3390/ani13020227

**Published:** 2023-01-07

**Authors:** Elsa K. Ludwig, Kallie J. Hobbs, Caroline A. McKinney-Aguirre, Liara M. Gonzalez

**Affiliations:** Department of Clinical Sciences, North Carolina State University, Raleigh, NC 27606, USA

**Keywords:** colic, intestine, ischemia, biomarker, acute phase protein, cytokine

## Abstract

**Simple Summary:**

Biomarkers are measurable substances within body tissues or fluids that allow for the identification of ongoing injury or disease. Colic secondary to gastrointestinal disease is one of the most frequent causes of morbidity and mortality in horses. Specifically, colic associated with intestinal ischemia is the most life-threatening variety of this disease. Optimization of biomarkers for the diagnosis of colic and identification of intestinal ischemia may expedite the diagnosis and management of this disease and thus help to alleviate this burden on the equid population. Lactate, and specifically the L isomer, is a commonly employed biomarker in colic evaluations. A variety of other biomarkers, however, have been preliminarily evaluated for equine colic. This paper reviews currently explored biomarkers in equine medicine for colic. Ultimately, based on this review, L-lactate continues to be the most reliable marker for intestinal ischemia during colic. However, further exploration of the biomarkers included here may eventually provide the key to accelerated identification, intervention, and thus better outcomes for horses suffering from intestinal ischemia.

**Abstract:**

Biomarkers are typically proteins, enzymes, or other molecular changes that are elevated or decreased in body fluids during the course of inflammation or disease. Biomarkers pose an extremely attractive tool for establishing diagnoses and prognoses of equine gastrointestinal colic, one of the most prevalent causes of morbidity and mortality in horses. This topic has received increasing attention because early diagnosis of some forms of severe colic, such as intestinal ischemia, would create opportunities for rapid interventions that would likely improve case outcomes. This review explores biomarkers currently used in equine medicine for colic, including acute phase proteins, proinflammatory cytokines, markers of endotoxemia, and tissue injury metabolites. To date, no single biomarker has been identified that is perfectly sensitive and specific for intestinal ischemia; however, L-lactate has been proven to be a very functional and highly utilized diagnostic tool. However, further exploration of other biomarkers discussed in this review may provide the key to accelerated identification, intervention, and better outcomes for horses suffering from severe colic.

## 1. Introduction

Colic is one of the most frequent causes of equine morbidity and mortality, affecting 3.5–11% of horses each year [1]. Approximately 11% of affected horses will die from colic [2]. The most common cause of colic-related death is attributed to advanced ischemic damage to the intestine, a sequela of strangulating obstructions that affect approximately 21% of colicking horses referred to veterinary hospitals [2,3,4]. The timely diagnosis and treatment of intestinal ischemia is vital for decreasing patient morbidity and mortality; therefore, the ability to quickly and accurately diagnose and treat intestinal ischemia is critical for improving patient survival [5,6,7].

Intestinal ischemia, inflammation, or injury, produce biomarkers that are disease-associated molecular changes within bodily tissues and fluids. In humans, hundreds of biomarkers have been investigated and used as diagnostic indicators for a multitude of diseases [8]. Commonly used biomarkers for the diagnosis of human intestinal injury and ischemia include lactate, procalcitonin (PCT), ischemia-modified albumin (IMA), endothelin-1, intestinal fatty acid binding protein (I-FABP), a-Glutathione S-transferase (a-GST), interleukin-6 (IL-6), C-reactive protein (CRP), and serum amyloid A (SAA) [7,9]. Of these, IMA and I-FABP have recently been found to be sensitive indicators of gastrointestinal tract ischemia [10,11]. Unfortunately, no single human biomarker has been found to be completely accurate for disease diagnosis [12,13].

Although intestinal biomarker research has been primarily focused on humans, there are a limited number of previously published papers on equine intestinal biomarkers. In equine medicine, veterinarians are currently limited to the use of lactate, creatine kinase (CK), and albumin to aid in the determination of intestinal ischemia in colicking horses. Although these markers are useful in establishing surgical vs. non-surgical intestinal lesions, they are not 100% predictive; thus, further investigation into biomarkers for intestinal injury and ischemia is warranted [14]. The most effective diagnostic biomarker should be highly sensitive and specific for intestinal ischemia and ideally measurable peripherally in the blood as well as within the abdominal fluid. Ischemia initially affects the intestinal mucosa; thus, mucosa-derived biomarkers may provide earlier signs of ischemic injury; however, other intestine-specific biomarkers, alone or in combination, could be diagnostic [15]. The aim of this study was to review the literature regarding equine biomarkers for colic as well as commonly studied human intestinal injury biomarkers. 

## 2. The Acute Phase Response

The acute phase response (APR) is a crucial component of the innate immune system and is induced by infection, inflammation, or injury [16]. Components of this response, such as proinflammatory cytokines and acute phase proteins (APPs), have therefore been used to determine the prognosis and diagnosis of intestinal disease. The APR functions to remove the inciting cause of inflammation, promote healing, and restore normal physiological function [16,17]. This response is activated when injured cells release alarm molecules, such as reactive oxygen species, arachidonic acid metabolites, and products of oxidative stress [17]. These alarm molecules activate cells that produce inflammatory mediators such as cytokines [16,17].

Proinflammatory cytokines are mediators that play an important role in the response to injury through the production of more cytokines and other inflammatory mediators. This results in the production of APPs that are required for immune system modulation, complement activation, protein transport, and tissue protection and healing [18,19,20,21]. The major cytokines involved in the APR include interleukin-6 (IL-6), interleukin-1β (IL-1β), and tumor necrosis factor-α (TNF-α), and these cytokines produce the clinical signs associated with inflammation or infection stimulate other cells in the APR cascade, and activate the production of APPs [16,17]. 

Research regarding the roles of different cytokines in intestinal health and injury is currently underway, and a better understanding of cytokine involvement in the inflammatory response will likely result in the development of cytokine-specific diagnostic and prognostic biomarkers [18]. Cytokine assays have been proposed for use in quantifying the systemic inflammatory response; however, cytokines may not be the ideal diagnostic biomarker due to their short half-lives [21,22]. The effect of gastrointestinal disease on equine cytokines has been investigated for IL-6, IL-1β, TNF-α, procalcitonin, and activin A, which are described in the subsequent sections.

Acute phase proteins are plasma glycoproteins that mediate the inflammatory response and modulate the immune response [16,19,23]. Acute-phase protein concentrations can either increase (positive APPs) or decrease (negative APPs) in response to inflammatory processes [16,17]. Serum amyloid A (SAA), haptoglobin, fibrinogen, and C-reactive protein (CRP) have all been explored as biomarkers to differentiate the cause of intestinal disease in the horse [16,17,24,25]. 

Overall, due to the high variability of the APR between individuals, many proinflammatory cytokines and APPs do not have well-established or validated diagnostic value ranges, and the degree of proinflammatory cytokine and APP responses to injury or inflammation is best determined via comparison to an individual’s own baseline values. Therefore, at this time, biomarkers of the APR are best utilized to help diagnose and prognosticate colic but cannot reliably be used to differentiate between causes of colic in horses. 

## 3. Proinflammatory Cytokines

### 3.1. Interleukin-6

Interleukin-6 is considered the primary cytokine stimulator of the APPs and is expressed in response to IL-1β and TNF-α [21,26]. In humans, there is strong evidence that IL-6 plays an integral role in intestinal inflammation, especially in cases of irritable bowel disease, enteritis, and colitis [27]. Clinical reports of IL-6 responses in humans with gastrointestinal disorders have found significantly higher blood IL-6 concentrations in ischemic versus non-ischemic intestinal disease, and IL-6 was found to be both sensitive and specific in a small population of patients with ischemic bowel [28,29]. 

Despite equine IL-6 being heavily researched for a multitude of diseases, there are limited studies with respect to the effect of intestinal diseases on serum and peritoneal fluid IL-6 concentrations. A 2009 study by Nieto et al. experimentally induced endotoxemia in horses by intravenous administration of lipopolysaccharide (LPS) and assessed the gene expression of a set of inflammatory cytokines, including IL-6 [26]. Interleukin-6 expression peaked in the blood at 90 min post-LPS administration and remained elevated for 3 h [26]. In 1999, Barton and Collatos evaluated the diagnostic and prognostic utility of measuring IL-6 levels in 155 horses that presented to a referral hospital for colic [30]. The authors reported that blood and peritoneal fluid IL-6 levels were significantly higher in horses with strangulating intestinal lesions compared to horses without strangulating lesions and that IL-6 levels were more frequently increased in the peritoneal fluid versus the serum [30]. Furthermore, IL-6 levels in the serum or peritoneal were correlated with mortality in the presenting cases [30]. This finding was in accordance with a 1995 study by Steverink et al., which assessed the concentrations of specific cytokines in 55 horses with colic [31]. Interleukin-6 concentrations were highest in horses affected with ischemic and inflammatory intestinal diseases, and IL-6 concentrations were predictive of poor outcomes [31]. Finally, IL-6 blood concentrations were found to be significantly greater in horses with *Clostridium difficile*-induced enterocolitis compared to healthy horses [32]. Based on these findings, measuring serum and peritoneal fluid IL-6 levels shows promise as a diagnostic and prognostic indicator for ischemic or inflammatory causes of colic.

### 3.2. Interleukin-1β

Interleukin-1β is a marker of acute inflammation often used for the evaluation of experimentally induced intestinal injury in rats [33,34,35]. Intestinal ischemia-reperfusion injury in rats resulted in significantly elevated serum IL-1β concentrations compared to control rats, and a similar finding was reported in the human intestine undergoing ischemia and reperfusion [34,35,36]. Mucosal IL-1β mRNA levels were elevated in human patients affected with eosinophilic colitis and Crohn’s disease, and elevated IL-1β mRNA levels indicated patients with early clinical relapse of Crohn’s disease [37,38]. Interestingly, in the previously described Nieto et al. study that evaluated the induction of inflammatory biomarkers in response to LPS administration, IL-1β gene expression peaked at 60 min post LPS injection, and serum IL-1β concentrations were found to be significantly higher in horses with *Clostridium difficile*-induced enterocolitis compared to healthy horses [26,32]. Colicking horses with intestinal strangulations were found to have significantly higher serum IL-1β concentrations when compared to a control group of healthy horses, however, there was no correlation between serum IL-1β concentrations and patient survival [39]. 

### 3.3. Tumor Necrosis Factor-α

Tumor necrosis factor-α (TNF-α) is a cytokine involved in cell signaling associated with the inflammatory response. In human and animal studies, peak serum TNF-α values correlate with peak levels of injury due to intestinal ischemia and elevated TNF-α concentrations have been found in patients with Crohn’s disease and inflammatory bowel disease [33,34,37,40]. In horses, TNF-α was the first identified cytokine measured in the circulation following the induction of endotoxemia using LPS, which ultimately induced the release of IL-6 and IL-1β.^26^ Several equine studies have evaluated the response of TNF-α to different causes of intestinal injury [30,31,32,41]. Overall, horses affected with strangulating intestinal lesions, enteritis, colitis, and ischemic/inflammatory lesions have significantly elevated TNF-α concentrations when compared to healthy horses or horses with non-strangulating and non-inflammatory forms of colic [30,31,32,41]. Furthermore, elevated serum TNF-α concentrations have been associated with high mortality rates in colicking horses [30,31,32,41].

### 3.4. Activin A

Activin A is rapidly elevated in the blood in response to inflammation and induces the release of other proinflammatory cytokines involved in the APR [42,43,44]. Serum activin A concentrations have been reported to be elevated in humans with inflammatory bowel disease; however, there are very limited equine activin A studies [45]. In a 2011 study by Forbes et al., activin A levels were evaluated in the serum of horses undergoing evaluation for acute abdominal disease [43]. The horses of this study were separated into three groups depending on their intestinal lesions, inflammatory, non-strangulating, or strangulating, and the serum activin A concentrations were compared between groups and control horses [43]. Compared to controls, serum activin A was significantly greater in horses with inflammatory or strangulating lesions; however, the authors state that both strangulating and non-strangulating intestinal lesions likely cause varying degrees of inflammation and therefore, activin A should not be relied on to differentiate between these causes of colic [43]. Based on these findings, Copas et al. further investigated the response of activin A to gastrointestinal inflammation by evaluating the differences in serum activin A concentrations between horses affected with equine grass sickness (EGS), healthy horses, unaffected horses co-grazing with EGS horses, and non-inflammatory colic cases [42]. Interestingly, the activin A levels of horses affected with EGS were not significantly different from any of the other groups, while the co-grazing horses’ levels were significantly greater than both normal horses and non-inflammatory colic cases [42]. The authors suggest that the co-grazing horses may have sub-clinical enteritis, indicating widespread exposure to the etiological agent of EGS [42]. Both equine studies conclude that activin A may have limited use as a diagnostic colic biomarker, as a multitude of intestinal diseases result in its’ elevation [42,43].

### 3.5. Procalcitonin 

Procalcitonin (PCT) is a peptide precursor to the hormone calcitonin, which is involved in calcium homeostasis. Proinflammatory cytokines such as TNF-α and IL-6 stimulate the secretion of PCT into the circulation [46,47,48]. Significantly elevated serum PCT levels have been successfully used in both humans and horses to diagnose sepsis and SIRS, as PCT concentrations have been shown to elevate rapidly in response to bacterial infection and endotoxemia [8,49,50]. Furthermore, PCT has been well-researched in human intestinal diseases and has been found to be predictive for intestinal ischemia, necrosis, the degree of intestinal injury, and patient prognosis [46,47,48]. These findings are similar in horses, where serum PCT levels were significantly higher in horses with colic; however, serum PCT levels were not shown to be effective in differentiating strangulating from non-strangulating intestinal lesions [39,50,51]. However, Kilcoyne et al. found that peritoneal fluid PCT levels are more sensitive for the diagnosis of intestinal ischemia than serum levels, likely due to local secretion from the intestines [51].

## 4. Acute Phase Proteins

### 4.1. Serum Amyloid A

Serum amyloid A, a major APP in horses, is present at very low concentrations in healthy horses and has a short half-life which makes it an ideal biomarker for monitoring ongoing inflammation and treatment response [24]. In horses, elevations in SAA concentrations have been associated with a variety of causes, including gastrointestinal disease, and the magnitude of change in SAA concentration varies depending on the inciting cause of inflammation [20,24,52,53]. In studies of colic, serum amyloid A has been evaluated for its ability to prognosticate colic outcome, identify the necessity of surgical intervention, diagnose early infection or post-celiotomy complications and differentiate between non-inflammatory causes of colic and equine grass sickness or colitis, and strangulating versus non-strangulating lesions [32,42,43,54,55,56,57,58,59,60,61,62]. In one study, colicking horses were reported to have significantly elevated serum and peritoneal fluid SAA concentrations compared to normal horses; however, the SAA concentrations in the peritoneal fluid were not greater than those of serum [42,63,64]. This finding is interesting as SAA can be locally produced in a multitude of tissues, including the intestine, and is often elevated at the site of injury [64,65]. A 2015 study by Pihl et al. evaluated the association of APPs with different equine intestinal disease durations and inciting causes such as simple obstructions, strangulating obstructions, or inflammatory diseases [52]. These authors found that both disease process and duration were significantly correlated with the concentration of serum and peritoneal fluid APPs and that SAA appeared to be the most clinically useful APP biomarker they investigated. In this study, serum SAA concentrations were most elevated in inflammatory diseases and in colic cases with durations greater than 5 h [52]. In the Copas et al. study, SAA concentrations were significantly elevated in inflammatory colics and in horses with EGS compared to co-grazers and healthy horses [42]. However, serum amyloid A concentrations were not significantly different between the EGS and inflammatory colic groups and therefore, the authors cautioned that SAA might not reliably differentiate between different causes of abdominal inflammation [42]. Serum SAA concentrations were also found to be significantly elevated in horses affected with acute colitis and *Clostridium difficile*-induced enterocolitis compared to normal horses and even horses with obstructive intestinal lesions [32,53,59]. Westerman et al. compared SAA concentrations between medical (displacements, impactions, or spasmodic colic) and surgical colics and found that elevated SAA concentrations were significantly associated with surgical colics and small intestinal obstructions [60]. Additionally, the authors reported that colicking horses with elevated SAA were more likely to have an overall poor prognosis [60]. However, the authors stipulated that many of those criteria overlapped as the majority of the obstructions in the study were strangulating and the horses presented with a prolonged duration of colic [60]. In contrast, when SAA concentrations were compared between horses affected with strangulating versus non-strangulating intestinal lesions and surgical versus non-surgical cases, Dondi et al. found no difference between the groups [66]. In all horses following colic surgery, SAA concentrations were significantly elevated for prolonged durations compared to baseline values and horses that underwent minor elective surgeries [55,58,62]. Post-colic surgery, the magnitude of SAA elevation was more significant in horses with post-operative complications such as colic, reflux, surgical site infection, diarrhea, and other causes of systemic inflammation [55,58]. Additionally, elevated SAA concentrations have been correlated with decreased patient survival, related to the duration and severity of the intestinal lesion [60,62,63]. Overall, while SAA is unable to definitely localize a colic lesion, it appears to be a helpful adjunct diagnostic for the prognostication of colic outcomes, with greater elevations associated with decreased survival. 

### 4.2. Haptoglobin

During injury and inflammation, haptoglobin binds free hemoglobin that is released from damaged red blood cells, thereby reducing oxidative damage and helping to prevent the loss of iron [17]. By complexing with hemoglobin and binding the iron that is required for bacterial growth, haptoglobin has a bacteriostatic effect [17]. Haptoglobin is a moderate equine APP and has been found to significantly elevate in response to intestinal inflammation in horses with multifactorial colic, such as those affected with concurrent *Clostridium difficile*-induced enterocolitis and equine grass sickness [17,23,32,54,67,68]. In 2013 and 2015, Pihl et al. compared serum and peritoneal fluid haptoglobin concentrations from healthy horses and those with colic [52,64]. Interestingly, these studies reported contradictory findings in haptoglobin concentration between peritoneal fluid and serum. Haptoglobin concentrations in the peritoneal fluid were significantly elevated in colicking horses compared to healthy horses. Furthermore, peritoneal fluid haptoglobin elevated more rapidly (by 12–24 h) in horses with strangulating intestinal lesions versus simple obstructions or inflammatory disease, as well as in horses with longer durations of colic [52]. In serum, however, haptoglobin levels were either decreased or unchanged in colicking horses compared to healthy reference horses [52,64]. Westerman et al. further evaluated the response of serum haptoglobin in colicking horses, comparing haptoglobin concentrations from healthy horses, medical colics, and surgical colics [60]. Similar to the findings from Pihl et al., serum haptoglobin concentrations were not significantly different between groups suggesting that serum haptoglobin may not be an ideal colic biomarker [52,60,64]. Westerman et al. stipulated that this may be due to the acute duration of colics included in the study, as haptoglobin takes 12-24 h to increase following inflammation or injury [60]. While of limited utility in acute disease, serum haptoglobin may be useful as a marker for chronic gastrointestinal inflammation or injury [23,52,60].

### 4.3. Fibrinogen

In inflammatory conditions, fibrinogen is involved in tissue repair and induces an intracellular signaling cascade that upregulates cellular phagocytosis, degranulation, and cytotoxicity [17]. Fibrinogen is one of the most commonly studied equine APPs and is frequently used as an indicator of systemic inflammation in horses, albeit an insensitive marker, as fibrinogen has a slow response time to inflammation, is consumed during coagulation, and has a wide reference range [17,58,60,69]. Inflammatory gastrointestinal diseases such as colitis, enteritis, peritonitis, and equine grass sickness cause similar, significant elevations in plasma and peritoneal fluid fibrinogen concentrations compared to healthy control horses, with peritoneal fluid changes occurring earlier than in blood [42,52,70]. Additionally, elevated plasma fibrinogen does not significantly differ between medical and surgical colics or specific intestinal lesions and does not correlate with the development of post-celiotomy complications [55,58,60]. However, a 2020 study by De Cozar et al. evaluated plasma fibrinogen concentrations before and after emergency colic surgery and found that horses presenting with elevated plasma fibrinogen and strangulating lesions were more likely to develop a post-operative complication [62]. This finding was possibly due to prolonged disease durations, correlating with the results of the 2015 Pihl et al. study [52,62]. Aside from elevated fibrinogen concentrations being associated with colic duration, fibrinogen may not be an ideal diagnostic biomarker for horses with colic due to its delayed response to inflammation and lack of sensitivity and specificity [17,52,55,58,60].

### 4.4. C-Reactive Protein

C-reactive proteins (CRP) are produced by the liver during the APR, following stimulation by IL-6, IL-1β, and TNF-α, and help bind damaged cells and enhance phagocytosis [71,72]. C-reactive protein is a highly sensitive marker for tissue injury and inflammation and is often used as a marker for sepsis, Crohn’s disease, and inflammatory bowel disease in humans and animals [22,68,72]. Additionally, the response of CRP to the treatment of gastrointestinal inflammation is used as an indicator of the effectiveness of therapy; CRP level decrease is associated with reduced intestinal inflammation [72]. In horses, CRP is a moderate APP, as it begins to increase approximately 3–5 days after the inflammatory stimulus, and elevated concentrations have been reported in horses with sepsis, colic, enteritis, and horses that underwent experimental jejunojejunostomies [17,68,73,74]. Given the delayed rise of this biomarker, CRP is not helpful in the identification of acute colic etiologies, like ischemic lesions, but may be useful for chronic inflammatory gastrointestinal conditions. 

## 5. Other Inflammatory Biomarkers

Beyond those already reviewed, inflammatory biomarkers previously evaluated for their ability to identify intestinal ischemia include intestinal fatty acid binding protein (I-FABP), matrix metalloproteinase-9 (MMP-9), hyaluronan, cell-free DNA (cfDNA), peritoneal bile acids, creatine kinase (CK), D-lactate, and L-lactate. Apart from well-studied L-lactate, many of the other inflammatory biomarkers listed have only been preliminarily explored in regard to colic. Of those included below, only lactate has been found to be specific for intestinal ischemia, with L-lactate proving to be the ideal biomarker. Table 1 summarizes the findings of currently available literature in the field of equine colic. 

### 5.1. Intestinal Fatty Acid Binding Protein

Intestinal fatty acid binding protein (I-FABP) is a cytosolic protein expressed by intestinal enterocytes located at the villus tips [12,28]. This protein is involved in the uptake and intracellular transport of fatty acids and is the only biomarker specific to the small intestine [28,75]. During homeostasis, I-FABP ranges from undetectable to low concentrations in the circulating blood, but when intestinal injury occurs, I-FABP is released into the peripheral bloodstream [12,28]. Many studies have found that I-FABP is significantly elevated in patients or animals with intestinal ischemia when compared to controls, and I-FABP has even been used to discern between a strangulating intestine and intestinal obstruction [28,75,76,77,78]. A 1993 study by Gollin et al. subjected rats to mesenteric ischemia by occluding the superior mesenteric artery for 30 min, 1 h, and 3 h, followed by reperfusion for up to 5 h [79]. Baseline I-FABP was <4.0 ng/mL and did not change in control animals throughout the duration of the experiment. However, in rats that underwent 30 min of ischemia and reperfusion, I-FABP rose significantly by 30 min of reperfusion, with the levels peaking at 1 h. When the rats were subjected to 1 h of ischemia, the I-FABP levels rose significantly within 15 min of reperfusion and peaked at 90 min. When the rats were allowed to recover, I-FABP returned to baseline levels 24 h after intestinal reperfusion. This study found I-FABP to be a sensitive and specific biomarker for ischemic intestinal injury [79]. 

In human models of intestinal ischemia and reperfusion, I-FABP has also been found to detect irreversible intestinal ischemia-reperfusion damage [77]. In a similar study design as that performed in rats, segments of human jejunum were exposed to 15, 30, or 60 min of ischemia and then allowed to reperfuse for 30 or 120 min [77]. Of note, the intestine that was subjected to ischemia was planned for removal for surgical reasons [77]. In this study, I-FABP levels were assessed in the patients’ blood and were found to be significantly elevated after a minimum of 30 min of ischemia. The authors also reported that there was a relationship between the amount of I-FABP released from enterocytes and the severity of the histologic mucosal injury. Based on this study, I-FABP can be used to differentiate between mild and reversible ischemic damage and severe and irreversible damage [77]. This finding may help clinicians determine if the degree of intestinal injury in a patient warrants surgical intervention, expediting treatment for intestinal ischemia. 

There is a paucity of equine literature on I-FABP and intestinal ischemia. Nieto et al. studied the I-FABP serum and peritoneal fluid levels in horses presenting with colic to a referral facility and found that high concentrations of I-FABP in peritoneal fluid correlated with non-survival, while plasma I-FABP concentrations correlated with the requirement of colic surgery [80]. While useful, peritoneal I-FABP was not perfectly predictive of surgical necessity and thus, combining evaluation with other biomarkers reviewed here is recommended to expand practitioners’ ability to diagnose and treat equine colic [80]. 

### 5.2. Matrix Metalloproteinase 9

Matrix metalloproteinases (MMPs) are a family of calcium-dependent zinc-containing proteases which play central roles in tissue morphogenesis, wound healing, and remodeling through mediating extracellular matrix turnover and recruiting inflammatory cells into the intestine [81]. These proteases are primarily produced by neutrophils following activation by proinflammatory mediators such as lipopolysaccharide (LPS), TNF-α, interleukin-8, and granulocyte-colony-stimulating factor [82]. As intestinal injury results in the breakdown of the epithelial barrier and increased permeability to LPS, investigators have previously queried how MMP production changes in intestinal diseases. In humans, increased MMP-9 has been associated with inflammatory bowel disease, sepsis, and exposure to pancreatic trypsin, as occurs in intestinal ischemia and reperfusion injury [81,83]. Of the over 20 MMPs which have been identified thus far, MMPs 2, 8, and 9 have been evaluated in equine colic-associated endotoxemia [82]. In this study, MMP-9 was specifically increased in colic. When comparing serum and peritoneal levels of these MMPs with sepsis score, peritoneal MMP-9 was found to hold promise as an indicator of the potential to develop sepsis in cases of colic [82]. Though MMP-9 concentration could not differentiate the type of colic lesion or degree of ischemia, further optimization of this marker may provide another tool for evaluating horses suffering from colic.

### 5.3. Hyaluronan

Hyaluronan is a glycosaminoglycan that is present in all extracellular matrices and functions as a mediator of inflammation by enhancing wound healing [84]. Two of the primary sources of hyaluronan secretion are the mesothelial cells of the abdomen and the vascular endothelium [85,86]. During times of vascular stress, such as during ischemia or sepsis, hyaluronan is shed into the circulation by damaged endothelial glycocalyx, and thus the quantification of systemic or local hyaluronan concentrations can be used to assess the health of the endothelial glycocalyx [85,86]. In humans, hyaluronan has been suggested to be a potential cause of intestinal disease, as the accumulation of endothelial-derived hyaluronan has been noted to drive the development of colitis [87]. Hyaluronan levels were evaluated in peritoneal fluid collected from a population of horses that presented to a referral hospital for colic and a control population of horses [84]. The peritoneal hyaluronan levels of the horses with colic were significantly elevated compared to the control horse levels; however, this study did not attempt to further correlate different causes of colic (i.e., impaction, intestinal ischemia, colon displacement, etc.) with the hyaluronan levels [84].

### 5.4. Cell-Free DNA

Cell-free DNA (cfDNA) increases as a result of cell death and has been identified as a plasma biomarker for ischemia, severe systemic inflammation, and mortality in human patients with gastrointestinal diseases [88,89]. Additionally, in canine studies, plasma cfDNA was significantly increased in dogs with gastric dilatation–volvulus [90]. In light of these trends in other species, cfDNA has been evaluated in the plasma of horses presenting to a referral center for emergency care, including colic [91]. While the evaluation of equine plasma posed additional challenges compared to other species, median DNA-extracted plasma cfDNA was significantly higher in horses presenting for emergency care as well as in a subgroup of horses presenting with colic compared to healthy controls [91]. Evaluations of the utility of cfDNA in diagnosing specific colic lesions are ongoing. While the requirements of DNA extraction increase the labor, resource, and time intensiveness of this biomarker, compared to evaluations in unprocessed plasma, with further evaluation, cfDNA may ultimately provide another method for the prognosticating outcome of colic cases. 

### 5.5. Peritoneal Bile Acids

Colic may result in the alteration of hepatic function and the diffusion of luminal content, including bile acids, through the damaged epithelium into the peritoneal space. Conversely, a primary hepatic injury may result in signs of colic. With this overlap of hepatic and gastrointestinal disease, researchers have previously evaluated how bile acid concentrations change in both disease states [92,93]. While increased plasma bile acid concentrations are known to be specific for hepatic insufficiency, peritoneal bile acid concentrations (PBAC) and their dynamics in colic and liver dysfunction were not explored until very recently [93]. This evaluation of PBAC in horses presenting with colic identified that elevations of peritoneal bile acids (>2.28 μmol/L) were associated with ischemic or inflammatory gastrointestinal lesions and non-survival [93]. Though researchers were able to establish a trend associating PBAC with colic and prognosis, because of the preliminary nature of these findings, additional evaluations are needed.

### 5.6. Creatine Kinase 

Creatine kinase (CK) plays a pivotal role in cellular energy homeostasis, particularly in tissues with highly dynamic energy demands, such as the intestine. In rodent models of intestinal ischemia, CK has been shown to increase in plasma [94]. Additionally, increases in peritoneal CK have been demonstrated in rabbit and equine models of strangulating intestinal obstructions [95,96]. When evaluated in horses presenting for colic, researchers identified that elevated peritoneal CK (>16 IU/L) was a highly sensitive marker of a strangulating lesion, even more sensitive, though less specific, than peritoneal lactate [97]. These findings suggest that measuring peritoneal CK may be a useful adjunct, especially in combination with the highly specific marker, lactate, to expedite the diagnosis and treatment of horses with strangulating intestinal lesions.

### 5.7. D-Lactate

D-lactate, the stereoisomer of the well-established colic biomarker L-lactate, may hold promise as a biomarker for equine colic as well. This form of lactate is produced by bacterial fermentation, as opposed to mammalian tissues, and can be produced by many of the bacterial species known to inhabit the equine intestine [98]. D-lactate elevations in human plasma have been demonstrated in the face of intestinal ischemia and septic shock [99,100]. Additionally, peritoneal D-lactate concentrations are considered a useful biomarker for human septic peritonitis [101]. Given that colic may incorporate elements of those syndromes and that it creates intestinal environments which favor bacterial proliferation or translocation, researchers have evaluated the utility of D-lactate in colic diagnostics and specifically differentiating ischemic versus non-ischemic lesions [102]. While plasma D-lactate concentrations did not correlate with colic status, peritoneal D-lactate concentrations (>116.6 μmol/L) had a high sensitivity (81%) and moderate specificity (65%) for differentiating ischemic versus non-ischemic colic lesions and correlated with peritoneal L-lactate concentrations [102,103]. Though further validation is needed, this work indicates that peritoneal D-lactate concentration may serve as an additional indicator of ongoing strangulating obstruction.

**Table 1 animals-13-00227-t001:** Biomarkers with single-mention use in equine literature.

Biomarker	Author, Year	Evaluated in Plasma and/or Peritoneal Fluid	Major Findings	Clinical Relevance
I-FABP	Nieto, 2005 [80]	Plasma and peritoneal fluid	Elevated plasma I-FABP concentrations correlated with the necessity for colic surgery, and elevated peritoneal I-FABP concentrations correlated with patient non-survival.	Plasma and peritoneal fluid I-FABP may be useful for the prediction of the necessity for colic surgery and patient survival in horses affected with colic.
MMP-9	Barton, 2021 [82]	Plasma and peritoneal fluid	Elevated peritoneal fluid MMP-9 concentrations correlated with the development of sepsis and endotoxemia in colic cases.	Peritoneal fluid MMP-9 concentrations are preferable over plasma for the identification of sepsis and endotoxemia in colicking horses.
Hyaluronan	Lillich, 2011 [84]	Plasma and peritoneal fluid	Significantly elevated peritoneal fluid hyaluronan concentrations were found in horses affected with colic versus healthy horses. Plasma hyaluronan levels were not significantly elevated in horses with colic.	Hyaluronan was locally produced in the abdomen of horses with colic and may be useful as a marker for ischemia.
Cell-free DNA	Bayless, 2022 [91]	Plasma	Plasma cfDNA concentrations were significantly higher in horses presenting for emergency care, including horses with colic, compared to healthy horses.	Plasma cfDNA may be a useful aid in the diagnosis of horses with colic.
Peritoneal bile acids	Rodríguez-Pozo, 2022 [93]	Plasma and peritoneal fluid	Elevated peritoneal bile acids concentrations correlated with ischemic or inflammatory intestinal lesions and patient non-survival. Plasma bile acid concentrations were not elevated in horses affected with colic.	Peritoneal fluid bile acids concentrations are preferable over plasma for the identification of ischemic or inflammatory intestinal lesions and patient non-survival in horses affected with colic.
Creatine Kinase	Kilcoyne, 2019 [97]	Plasma and peritoneal fluid	Horses with colic had higher plasma and peritoneal fluid CK concentrations than healthy horses. While both plasma and peritoneal fluid CK levels were more significantly elevated in horses with strangulating lesions versus non-strangulating lesions, peritoneal fluid CK was found to be more sensitive and specific than plasma for the identification of strangulating lesions.	Peritoneal fluid CK concentrations may be a useful adjunct to clinical case presentation and other intestinal biomarkers for the diagnosis of strangulation lesions in horses with colic.
D-Lactate	Yamout, 2011 [102]	Plasma and peritoneal fluid	Peritoneal fluid D-lactate levels were significantly elevated in horses with colic compared to normal horses, and peritoneal fluid D-lactate levels were significantly more elevated in strangulating versus non-strangulating lesions. While plasma D-lactate levels were elevated in colicking horses compared to control horses, there was no difference in concentrations between strangulating or non-strangulating colic.	Peritoneal fluid D-lactate may be more useful than plasma for the identification of strangulating lesions in colicking horses.

### 5.8. L-Lactate

In recent years, measurements of systemic and peritoneal L-lactate have become a staple in the diagnosis of equine colic. This type of lactate increases in the blood and peritoneal fluid secondary to increased anaerobic glycolysis due to poor tissue perfusion, which can occur in the ischemic intestine and collaterally impacted sections of the bowel as well. The general use of L-lactate in equine medicine, and its correlation to systemic and gastrointestinal disease severity, has been reviewed elsewhere, so this section will briefly review its application to colic [104]. When evaluated for use in equine patients, our search returned 28 papers that mentioned lactate as a biological marker in colic, with five papers being specific for ischemia in equine colic. Table 2 summarizes the findings of these five papers. It is well-accepted that plasma and peritoneal lactate should be below 2.0 mmol/L in normal horses [14]. In all types of colic, peritoneal fluid lactate is significantly higher in ischemic lesions compared to non-ischemic colic [6]. Indeed, peritoneal lactate is superior to blood lactate for earlier identification of intestinal ischemia prior to circulatory collapse [97,105]. Differentiating more specific types of colic, however, is more complicated as no difference in peritoneal fluid lactate was identified in comparisons of non-strangulating forms of colic, such as duodenitis-proximal jejunitis and strangulating small intestinal lesions [14]. Peritoneal fluid-to-blood lactate ratios are often able to add clarity, with ratios greater than or equal to two being consistent with a strangulating small intestinal lesion, though ratios of one or greater have also been reported to be consistent with ischemia [6,14,106]. Of note, although lactate appears to have a high sensitivity and specificity for the prediction of strangulating lesions, it is not 100% sensitive or specific in any reported study. However, the utility of lactate for predicting outcomes increases with serial measurements [56,106]. Despite limitations in sensitivity and specificity, L-lactate continues to be an incredibly useful tool for identifying colic and differentiating types of colic lesions.

## 6. Conclusions

Efforts to identify biomarkers useful in equine colic diagnosis have significantly gained momentum in the last 7 years. Though surgical exploration remains the only definitive diagnosis for most colic lesions, these biomarkers hold promise for identifying a highly sensitive and specific, accessible, and easily quantified model biomarker in the future. When evaluating efforts to identify novel biomarkers, considerations such as the number of horses enrolled, the complex etiologies of colic cases included, and the ultimate colic lesion will provide the necessary material for a thorough, multivariate evaluation of the performance of these variables [56]. Currently, L-lactate, and specifically its measurement in peritoneal fluid and compared to plasma concentrations, remains the most reliable biomarker for intestinal ischemia. This biomarker should be combined with other clinicopathological and physical exam parameters, as well as the patient’s pain score, to best evaluate colic severity and prognosis. Despite increased attention, there is a multitude of biomarkers currently used to diagnose intestinal ischemia in humans that have yet to be explored in the horse. Additionally, while most evaluations have focused on blood and peritoneal fluid measurement, other body fluids such as saliva, urine, and feces may provide useful sources of biomarkers as well. 

As the timely diagnosis and treatment of intestinal ischemia is vital for decreasing patient morbidity and mortality, the use of point-of-care, stall-side tests is warranted to immediately determine biomarker levels. Unfortunately, the majority of the intestinal injury biomarker tests available are enzyme-linked immunosorbent assays (ELISA), which usually must be performed in an in-house laboratory and can take several hours to obtain results. Most equine practitioners must ship biological samples to diagnostic laboratories and therefore do not receive results for several days. At this time, only SAA and L-lactate levels can be measured using point-of-care, stall-side tests, however with expanding interest and advancing technology for point-of-care analysis, the potential for identifying an ideal colic biomarker and thus expediting the diagnosis and care of equids with colic, remains optimistic.

## Figures and Tables

**Table 2 animals-13-00227-t002:** Summary of five original research papers describing the use of L-lactate in the equine patient with colic.

Author, Year	Evaluated in Blood/Plasma and/or Peritoneal Fluid	Major Findings	Clinical Relevance
Shearer, 2018 [14]	Blood and peritoneal fluid	No significant difference was found in blood and peritoneal fluid L-lactate values from horses with non-strangulating and strangulating intestinal lesions. However, the peritoneal fluid L-lactate to blood lactate ratio was significantly elevated in horses with strangulating intestinal lesions.	The peritoneal fluid L-lactate to blood L-lactate ratio may help differentiate strangulating from non-strangulating intestinal lesions.
Kilcoyne, 2019 [97]	Plasma and peritoneal fluid	Peritoneal L-lactate of 3.75 mmol/L was highly specific (92%) and moderately sensitive (81%) for predicting a strangulating intestinal lesion in colicking horses.	Elevated peritoneal fluid L-lactate is a strong indicator of intestinal ischemia.
Delesalle, 2007 [105]	Plasma and peritoneal fluid	Significantly elevated blood and peritoneal fluid L-lactate values were found in horses with strangulating lesions versus non-strangulating lesions and in horses that required colic surgery compared to horses medically managed. Peritoneal fluid L-lactate was significantly more elevated than blood L-lactate in horses that did not survive.	While both blood and peritoneal fluid L-lactates can be prognostic indicators in horses with colic, peritoneal fluid L-lactate is more predictive for intestinal ischemia and case outcome than blood L-lactate.
Latson, 2005 [6]	Plasma and peritoneal fluid	Horses affected with strangulating or non-strangulating intestinal lesions had significantly elevated peritoneal fluid and plasma L-lactate levels compared to normal horses. L-lactate levels were more significantly elevated in peritoneal fluid than plasma in horses with intestinal strangulation and ischemia.	Peritoneal fluid L-lactate is a better predictor of strangulating lesions and intestinal ischemia than blood L-lactate levels.
Peloso, 2012 [106]	Blood and peritoneal fluid	Peritoneal fluid L-lactate levels were significantly higher at admission to the hospital and after 6 h of hospitalization in horses with strangulating lesions than non-strangulating lesions. Horses with strangulating lesions had significantly elevated peritoneal fluid-to-blood L-lactate ratios compared to horses with non-strangulating lesions. Peritoneal fluid L-lactate >4 mmol/L and an increase in peritoneal fluid L-lactate levels over time were predictive for strangulating intestinal lesions.	Peritoneal fluid L-lactate levels can be used to help differentiate between strangulating and non-strangulating intestinal lesions in horses with colic.

## Data Availability

No new data was created for this publication.

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
