# Peer review of "Biomarkers of Intestinal Injury in Colic"

_animals, 2023, doi:10.3390/ani13020227_

Round 1
Reviewer 1 Report
Thank you so much for this comprehensive and extremely well-written review of literature that is incredibly pertinent to the study of colic in today's research and clinical arena. I found the paper to be well-organized, factually correct, and easy to follow. The tables will make a handy guide for inquiry into primary literature. I only have 2 minor points:
1. The segment on hyaluronan (lines 392-403) seems to suggest (at least to me) that it is a GAG specific to mesothelial cells of the abdomen. A sentence or two on hyaluronan shedding from the endothelium may be warranted, as this is another potential reason for increased blood levels of hyaluronan in horses presenting with signs of sepsis related or unrelated to abdominal disease. The paper below helped me review this topic, but the very knowledgeable authors of this paper under review may know of other primary or equine literature.
Dogné, S., & Flamion, B. (2020). Endothelial glycocalyx impairment in disease: focus on hyaluronan shedding. The American journal of pathology, 190(4), 768-780.
2. The sentence on line 494-497 regarding lactate levels seems to suggest that ischemic lesions have peritoneal lactate levels over 8.45 mmol/L, which may mislead a reader to think that any lesser lactate would suggest a non-ischemic pathology. I much prefer the summary in Table 2; removing the numbers (8.45 vs 2.09) in the paragraph may be all that is needed.
Author Response
The authors sincerely thank the reviewers for their suggestions regarding the manuscript titled “A Review of Intestinal Injury Biomarkers in Colicking Horses”. The edits we have made based on the reviewers’ comments have significantly strengthened the manuscript. Below is a summary of the changes made to the revised manuscript. Reviewer comments are bolded and author responses follow each comment. Within the revised manuscript the edits are highlighted in yellow. Please let us know if you have further suggestions. Thank you!
- The segment on hyaluronan (lines 392-403) seems to suggest (at least to me) that it is a GAG specific to mesothelial cells of the abdomen. A sentence or two on hyaluronan shedding from the endothelium may be warranted, as this is another potential reason for increased blood levels of hyaluronan in horses presenting with signs of sepsis related or unrelated to abdominal disease. The paper below helped me review this topic, but the very knowledgeable authors of this paper under review may know of other primary or equine literature. (Dogné, S., & Flamion, B. (2020). Endothelial glycocalyx impairment in disease: focus on hyaluronan shedding. The American journal of pathology, 190(4), 768-780.)
Thank you for this suggestion. The authors agree with this observation and appreciate the citation provided. The hyaluronan section has been updated to reflect the endothelium’s involvement in the production of hyaluronan (lines 381-386) and the following references have been added:
- Dogné, Sophie, and Bruno Flamion. "Endothelial glycocalyx impairment in disease: focus on hyaluronan shedding." The American journal of pathology 190.4 (2020): 768-780.
- Lennon, Frances E., and Patrick A. Singleton. "Hyaluronan regulation of vascular integrity." American journal of cardiovascular disease 1.3 (2011): 200.
- The sentence on line 494-497 regarding lactate levels seems to suggest that ischemic lesions have peritoneal lactate levels over 8.45 mmol/L, which may mislead a reader to think that any lesser lactate would suggest a non-ischemic pathology. I much prefer the summary in Table 2; removing the numbers (8.45 vs 2.09) in the paragraph may be all that is needed.
The authors agree with the reviewer’s observation that this may be a misleading statement. The value for lactate has been removed from the paper (lines 463-464).
Reviewer 2 Report
Dear authors:
The paper "biomarkers of intestinal injury in Colic" is a comprehensive and well written review of studied biomarkers used to identify severe colic in horses. It would be interesting to add to the text more clearly if a commercially available test for clinical use exists for the different biomarkers.
Furthermore, several biomarkers increase with ischaemia but also with inflammation and it should be more clearly stated, that those markers do not help so much for clinical cases, as it is particularly challenging to differentiate small intestinal strangulating obstruction from proximal enteritis.
The tables at the end of the document add a good overview to the text.
Please find in the following some specific comments:
Line 21
Bodily ? fluid à I do not know this expression, but I’m not native English speaking
Line 102-109
Ischaemia and inflammation are not easily identified with APR or other markers, but this point is important, as some pathologies in horses like proximal enteritis and strangulating obstruction of small intestine are not easy to distinguish, one case needs quick surgery whereas the other can be managed medically à you should better focus on this point when you speak about the APP
Line 122-141
Same comment for the IL-6 chapter on colic in horses, increase with ischaemia but also with inflammatory bowel disease
IL-1β same problem, increase in inflammatory and ischaemic bowel disease, no way to differentiate
TNF-α again cannot differentiate between inflammatory and ischaemic bowel disease
SAA
Cannot localize the lesion but also not possible to differentiate between surgical and medical lesion as colitis will also have increased values,
Line 300-304
Finding of de Cozar cannot be be confirmed by Pihl, the study of Pihl is 5 years earlier than de Cozar, please change phrasing
C-reactive Protein
Line 314 citation 22 and 68 not adequate for the statement (do not treat human disease but veterinary, there a lot of publications about C-reactive protein in humans (e.g. Sproston NR, Ashworth JJ. Role of C-Reactive Protein at Sites of Inflammation and Infection. Front Immunol. 2018;9:754. Published 2018 Apr 13. doi:10.3389/fimmu.2018.00754 and the references in this paper)
For all the following biomarkers, it should be interesting to the reader to add the information if the measurement is available for clinicians (point-of-care) (I-FABP, MMP9, Hyaluronan, cell-free DNA, peritoneal bile acids, D-lactate)
Line 337 sentence construction: concentrations are undetectable or too low or undetectably low ?
Author Response
The authors sincerely thank the reviewers for their suggestions regarding the manuscript titled “A Review of Intestinal Injury Biomarkers in Colicking Horses”. The edits we have made based on the reviewers’ comments have significantly strengthened the manuscript. Below is a summary of the changes made to the revised manuscript. Reviewer comments are bolded and author responses follow each comment. Within the revised manuscript the edits are highlighted in yellow. Please let us know if you have further suggestions. Thank you!
- Line 21: Bodily ? fluid à I do not know this expression, but I’m not native English speaking
The term “bodily fluids” means the same as “body fluids”, therefore the authors replaced “bodily” with “body” (line 21) for ease of understanding.
-
- Line 102-109: Ischaemia and inflammation are not easily identified with APR or other markers, but this point is important, as some pathologies in horses like proximal enteritis and strangulating obstruction of small intestine are not easy to distinguish, one case needs quick surgery whereas the other can be managed medically à you should better focus on this point when you speak about the APP
- Lines 122-141:
- Same comment for the IL-6 chapter on colic in horses, increase with ischaemia but also with inflammatory bowel disease
- IL-1β same problem, increase in inflammatory and ischaemic bowel disease, no way to differentiate
- TNF-α again cannot differentiate between inflammatory and ischaemic bowel disease
- SAA: Cannot localize the lesion but also not possible to differentiate between surgical and medical lesion as colitis will also have increased values
The authors thank the reviewer for these suggestions. We further expanded upon our original statement that there is significant variability of the APR between individuals to clarify that APR biomarkers cannot reliably differentiate between causes of colic as follows (lines 102-107): “Overall, due to the high variability of the APR between individuals, many proinflammatory cytokines and APPs do not have well established or validated diagnostic value ranges, and the degree of proinflammatory cytokine and APP responses to injury or inflammation is best determined via comparison to an individual’s own baseline values. Therefore, at this time, biomarkers of the APR are best utilized to help diagnose and prognosticate colic but cannot reliably be used to differentiate between causes of colic in horses.”
- Line 300-304: Finding of de Cozar cannot be be confirmed by Pihl, the study of Pihl is 5 years earlier than de Cozar, please change phrasing
Thank you for catching this error. The wording of the sentence (lines 292-293) has been changed to the following: “This finding was possibly due to prolonged disease durations, correlating with the results of the 2015 Pihl et al. study.”
- Line 314 citation 22 and 68 not adequate for the statement (do not treat human disease but veterinary, there a lot of publications about C-reactive protein in humans (e.g. Sproston NR, Ashworth JJ. Role of C-Reactive Protein at Sites of Inflammation and Infection. Front Immunol. 2018;9:754. Published 2018 Apr 13. doi:10.3389/fimmu.2018.00754 and the references in this paper)
Thank you for catching this error. We added “and animals” to the sentence so it is written as follows (lines 300-303): “C-reactive protein is a highly sensitive marker for tissue injury and inflammation and is often used as a marker for sepsis, Crohn’s disease, and inflammatory bowel disease in humans and animals”. References #22 and 68 do include discussion of both animal and human use of different biomarkers and thus we feel they appropriately support our statement.
- For all the following biomarkers, it should be interesting to the reader to add the information if the measurement is available for clinicians (point-of-care) (I-FABP, MMP9, Hyaluronan, cell-free DNA, peritoneal bile acids, D-lactate)
This is an excellent suggestion. The authors added a discussion section regarding the current availability of biomarker assays (lines 498-507): “As the timely diagnosis and treatment of intestinal ischemia is vital for decreasing patient morbidity and mortality, the use of a point-of-care, stall-side tests are warranted to immediately determine biomarker levels. Unfortunately, the majority of the intestinal injury biomarker tests available are enzyme-linked immunosorbent assays (ELISA), which usually must be performed in an in-house laboratory and can take several hours to obtain results. Most equine practitioners must ship biological samples to diagnostic laboratories and therefore do not receive results for several days. At this time only SAA and L-lactate levels can be measured using point-of-care, stall-side tests, however with expanding interest and advancing technology for point-of-care analysis, the potential for identifying an ideal colic biomarker and thus expediting the diagnosis and care of equids with colic, remains optimistic.”
- Line 337 sentence construction: concentrations are undetectable or too low or undetectably low?
The sentence has been updated to read as follows (line 325-327): “During homeostasis, I-FABP ranges from undetectable to low concentrations in the circulating blood, but when intestinal injury occurs, I-FABP is released into the peripheral bloodstream”.